# Does the New FIGO 2018 Staging System Allow Better Prognostic Differentiation in Early Stage Cervical Cancer? A Dutch Nationwide Cohort Study

**DOI:** 10.3390/cancers14133140

**Published:** 2022-06-27

**Authors:** Mieke L. G. Ten Eikelder, Floor Hinten, Anke Smits, Maaike A. Van der Aa, Ruud L. M. Bekkers, Joanna IntHout, Hans H. B. Wenzel, Petra L. M. Zusterzeel

**Affiliations:** 1Department of Gynecological Oncology, Radboud University Medical Center, 6525 GA Nijmegen, The Netherlands; floor.hinten@radboudumc.nl (F.H.); anke.smits@radboudumc.nl (A.S.); petra.zusterzeel@radboudumc.nl (P.L.M.Z.); 2Department of Research and Development, Netherlands Comprehensive Cancer Organization, 3501 DB Utrecht, The Netherlands; m.vanderaa@iknl.nl (M.A.V.d.A.); h.wenzel@iknl.nl (H.H.B.W.); 3Department of Obstetrics and Gynecology, GROW School for Oncology and Reproduction, Maastricht University Medical Center, 6200 MD Maastricht, The Netherlands; ruud.bekkers@catharinaziekenhuis.nl; 4Department of Obstetrics and Gynecology, Catharina Hospital, 5602 ZA Eindhoven, The Netherlands; 5Department for Health Evidence, Radboud University Medical Center, 6500 HB Nijmegen, The Netherlands; joanna.inthout@radboudumc.nl

**Keywords:** early-stage cervical cancer, FIGO staging 2018, overall survival, recurrence free survival, risk factors

## Abstract

**Simple Summary:**

The introduction of a revised staging system (FIGO 2018 staging system) for cervical cancer has led to a significant change in stage allocation for patients with early stage disease. It remains unclear how this change should be translated into treatment options, including less extensive surgery. With this Dutch national study we evaluated whether the revised staging system resulted in a more accurate prediction of overall and recurrence free survival compared to the previous FIGO 2009 staging system. In addition, we assessed other factors which may help the paradigm of treatment. We concluded that the revised FIGO 2018 staging system gives a more precise indication of survival outcomes of women with early stage cervical cancer. In addition, we believe that aside from stage, tumor characteristics, such as LVSI, and depth of invasion should be considered when offering patients less radical or less extensive treatment.

**Abstract:**

The FIGO 2018 staging system was introduced to allow better prognostic differentiation in cervical cancer, causing considerable stage migration and affecting treatment options. We evaluated the accuracy of the FIGO 2018 staging in predicting recurrence free (RFS) and overall survival (OS) compared to FIGO 2009 staging in clinically early stage cervical cancer. We conducted a nationwide retrospective cohort study, including 2264 patients with preoperative FIGO (2009) IA1, IA2 and IB1 cervical cancer between 2007–2017. Kaplan–Meier analyses were used to assess survival outcomes. Logistic regression was used to assess risk factors for lymph node metastasis and parametrial invasion. Stage migration occurred in 48% (22% down-staged, 26% up-staged). Survival data of patients down-staged from IB to IA1/2 disease were comparable with FIGO 2009 IA1/2 and better than patients remaining stage IB1. LVSI, invasion depth and parametrial invasion were risk factors for lymph node metastases. LVSI, grade and age were associated with parametrial invasion. In conclusion, the FIGO 2018 staging system accurately reflects prognosis in early stage cervical cancer and is therefore more suitable than the FIGO 2009 staging. However subdivision in IA1 or IA2 based on presence or absence of LVSI instead of depth of invasion would have improved accuracy. For patients down-staged to IA1/2, less radical surgery seems appropriate, although LVSI and histology should be considered when determining the treatment plan.

## 1. Introduction

Initially, the International Federation of Gynecology and Obstetrics (FIGO) staging of cervical cancer was primarily based on clinical examination [1]. In 2018, the FIGO staging classification was revised to maintain applicability worldwide by incorporating imaging modality and pathological characteristics [2]. The revised FIGO staging has resulted in several key changes for early stage cervical cancer. Firstly, women with microscopic tumor depth of invasion of ≤5 mm are allocated to stage IA disease irrespective of lateral extent. In addition, for stage IB disease an additional cutoff of 2 cm was introduced, resulting in three substages: stage IB1 (<2 cm), stage 1B2 (2–4 cm) and stage IB3 (≥4 cm). Furthermore, radiological or pathological nodal involvement resulted in upstaging to stage IIIC disease [2] (Table 1).

The revised staging was proposed to allow better differentiation of prognostic outcomes and facilitate better clinical management [2]. However, this revision caused a stage migration for a large proportion of women with early stage cervical cancer [3]. Women previously staged as IB1 stage disease based on lateral extent more than 7 mm with depth of invasion < 5 mm, are now considered FIGO stage 2018 IA disease and are possibly eligible for less radical surgery. Evidence regarding survival outcomes is, however, limited for this newly defined group. In addition, the revised stage IB1 with tumors < 2 cm has created a new group of women that may benefit from a less radical approach and less extensive surgery. However, clinical trials, such as the SHAPE trial (NCT01658930), are still awaited to further guide the paradigm of treatment in patients with tumors < 2 cm horizontal width [4].

Matsuo et al. validated the revised staging FIGO 2018 for stage IB and IIIC disease specifically, but further large cohort studies assessing the prognostic significance of the revised FIGO are lacking [3,5]. In addition, the new FIGO staging system, in contrast to FIGO treatment guidelines, does not incorporate other known prognostic factors, such as histopathological factors and lymph-vascular space invasion (LVSI), and therefore may not fully reflect risk groups [2,6,7].

With this Dutch nationwide cohort study, we aimed to evaluate the accuracy of the clinical FIGO 2018 staging in predicting the recurrence free survival (RFS) and overall survival (OS) compared to the FIGO 2009 staging for women with clinically early stage cervical cancer. Within these new staging groups, risk factors for lymph node metastasis and parametrial invasion were assessed to identify women who may benefit from less extensive surgery.

## 2. Methods

### 2.1. Study Design and Population

For this nationwide retrospective cohort study, we used data from the Netherlands Cancer Registry, a population-based registry with coverage of all newly diagnosed malignancies in the Netherlands since 1989. All patients newly diagnosed with FIGO (2009) IA1, IA2 and IB1 cervical cancer registered in the Netherlands Cancer Registry between January 2007 and December 2017 were included. Other histology than squamous-, adeno- and adenosquamous cell carcinomas were excluded. Furthermore, patients were excluded if they received neoadjuvant chemotherapy, were not treated with primary surgery, or if tumor characteristics, including both invasion depth and lateral extension, were missing.

### 2.2. Data Collection and Outcome Measures

Baseline-, clinical-, pathological- and treatment-related data were extracted from the Netherlands Cancer Registry, including age at diagnosis, body mass index (BMI), preoperative FIGO 2009 stage, type of surgery, histological subtype, differentiation grade, depth of invasion, linear extension, LVSI, lymph node status, parametrial invasion, disease recurrence and follow up. LVSI was considered positive when found either in pre-operative and/or in post-operative tissue, LVSI. Disease recurrence was defined as a histologically proven recurrence at least six months after initial treatment. Type of recurrence (local, locoregional or distant) was based on pathological findings. In patients with both locoregional and distant recurrence, patients were categorized as distant recurrences. Vital status and date of death were obtained by linking the registry to the municipal basic administration. We reallocated women from presurgical FIGO 2009 stage to presurgical FIGO 2018 stages, according to Table 1 [2], not adjusting for the final pathological findings after surgery.

### 2.3. Statistical Analyses

Continuous variables were presented as means with standard deviations or medians and interquartile range, as appropriate. Categorical outcomes were presented as frequencies and proportions. Demographic and clinical data were compared using Chi-square test for categorical data and one-way ANOVA for continuous data. The Kaplan–Meier method was applied to estimate recurrence-free and overall survival. Cox proportional regression with Hazard’s ratio were used to estimate differences in recurrence free and overall survival in different FIGO stages. Univariable logistic regression was used to assess the association for lymph node metastasis and parametrial invasion with clinical and histopathological characteristics. Multivariable logistic regression with forward selection procedure was used to identify those variables that independently contributed to the predicted risk of lymph node metastases, parametrial invasion and recurrence. The odds ratios with the 95% confidence intervals (CI) are presented. Statistical tests were two-tailed and considered significant at *p* < 0.05. Data were analyzed using IBM SPSS Statistics for Windows, version 25 (IBM Corp., Armonk, NY, USA).

## 3. Results

A total of 6605 patients were diagnosed with cervical cancer between January 2007 and December 2017 in the Netherlands. Of them, 2879 were diagnosed with FIGO 2009 stage IA1, IA2 or IB1; 52 were excluded because of non-squamous or non-adeno(squamous) histology, 563 were excluded because of unknown depth of invasion and unknown linear extension, resulting in a study population of 2264 patients (Figure 1).

According to the FIGO 2009 stage distribution, 35.6% (N = 806) of the patients were diagnosed in stage IA1, 4.0% (N = 90) in stage IA2 and 60.4% (N = 1368) in stage IB1. Fifty-two percent of patients underwent a radical hysterectomy, 21.9% a simple hysterectomy and 25.8% had fertility-preserving surgery. Baseline and clinical characteristics, including the preoperative FIGO stage, are presented in Table 2. Median age at diagnosis was 40 years (interquartile range 34–47) and squamous cell carcinoma (SCC) was the most prevalent histological subtype (Table 2).

After reallocation to preoperative FIGO 2018 stage, the stage of disease changed in almost half of the patients with early stage disease (N = 1096). Twenty-six percent of patients were up-staged from FIGO 2009 IB1 to FIGO 2018 IB2; whereas 22.1% of patients were down-staged from macro-invasive (IB1) towards micro-invasive disease (IA1 or IA2), as is illustrated in Figure 2. Baseline and clinical characteristics of patients who were down-staged (IB1 to IA1 and IA2) or up-staged (IB1 to 1B2) and patients who remained the same stage (IA1 and IA2, and IB1 respectively) are presented in Table 2.

### 3.1. Survival and Recurrence Analyses

Follow up was at least three years after surgery, with a median follow up of 83 months (IQR 57–111 months). Within follow up, 151 (6.7%) patients developed a recurrence; 3.4% local, 1.0% locoregional and 2.3% distant (with or without locoregional), with a median time to recurrence of 22 months. Recurrences occurred significantly more often in patients with macro-invasive, thus the depth of invasion > 5 mm, (FIGO 2018) disease than in women with FIGO 2018 micro-invasive disease (HR 3.85, 95% CI 2.85–5.20, *p* < 0.01). Women who were up-staged from IB1 to IB2 had significantly more recurrences than patients with a tumor less than 2 cm who remained IB1 (HR 2.02, 95% CI 1.32–3.10, *p* < 0.01) (Figure 3B).

Patients who were down-staged from macro-invasive disease to micro-invasive disease (from IB1 to IA1/2) had significantly more often recurrences than patients who had IA1/2 less than 7 mm (5.6% versus 2.5%; HR 2.21 95% CI 1.34–3.65 *p* < 0.01). The recurrence rate in these newly staged IA1/2 was similar to that in patients who remained IB1. Remarkably, OS did not significantly differ between women who had micro-invasive disease in FIGO 2009 as compared to women who became micro-invasive in the new FIGO (HR 1.48 95% CI 0.74–2.98; *p* = 0.26) (Figure 3A).

One hundred and seven patients (4.7%) had died of disease. Patients who were down-staged to micro-invasive disease had a better OS than women who remained macro-invasive IB1 in the FIGO 2018 staging (HR 2.54, 95% CI 1.22–5.29; *p* = 0.01). Ten years OS was 98.8% for IA1/2 < 7 mm, 97.4% for IA1/2 > 7 mm, 94.5% for FIGO 2018 IB1 and 89.2% for FIGO 2018 IB2, respectively.

### 3.2. Parametrial Invasion

A total of 1329 patients had radical surgery, either a radical hysterectomy or trachelectomy. Parametrial invasion was found in 36 patients: 23 (3.9%) in preoperative FIGO 2018 IB2, 11 (4.0%) in FIGO IB1 and two (0.4%) in preoperative FIGO 2018 IA2 (Table 2). The latter two patients were down-staged from FIGO 2009 macro-invasive disease towards micro-invasive disease. Both women had squamous cell carcinoma (SCC), LVSI present, depth of invasion < 3 mm, width > 7 mm and both underwent a radical hysterectomy. Parametrial invasion was significantly associated with age, tumor histology and differentiation, and LVSI and invasion depth (Table 3). After multivariable analysis, the presence of LVSI (OR 5.2 CI: 1.7–16.0) and high differentiation grade (OR 3.0, 95% CI: 1.3–6.8) remained significantly associated with parametrial invasion.

### 3.3. Lymph Node Involvement

In 1448 patients a lymphadenectomy was performed and 13.3% (N = 193) of patients showed lymph node involvement (LNM). Lymph node involvement according to histology, LVSI and FIGO stage 2018 are illustrated in Figure 4. The risk of lymph node involvement for FIGO stage 2009 IB1 -> 2018 IA1/2 was 4.8% (24/501), and this was significantly higher than FIGO stage 2009 IA1/2 with LVSI (1 LNM in 80 patients, 1.25%, *p* < 0.05). Furthermore, the risk of lymph node involvement for FIGO stage 2009 IB1 > 2018 IA1/2 was significantly higher when LVSI was present compared to negative LVSI 12.7% (N = 16/126) versus 2.1% (N = 6/292), *p* < 0.05). There was no difference for risk of lymph node metastases in this group for <3 mm versus 3–5 mm invasion (3.4% versus 6.0%, *p* = 0.21). As shown in Figure 3C, patients with IA1 or IA2 with LVSI were significantly more prone to LNM than patients with IA1 or IA2 without LVSI with a HR 3.36 95% CI 1.33–8.64; *p* < 0.01.

Risk factors for lymph node metastasis identified for all patients who underwent a lymphadenectomy were tumor grade, presence of LVSI, tumor size (both depth of invasion and horizontal width) and the presence of parametrium invasion (Table 4). For invasion depth, a depth of >5 mm was significantly associated with increased incidence of lymph node metastasis; there was no difference between <3 mm and 3–5 mm invasion depth (*p* = 0.12).

Notably, histology (adeno(squamous) versus squamous) was not significantly different in predicting lymph node metastasis. After multivariable analyses, the presence of LVSI (OR 6.1, 95% CI: 3.8–9.9), parametrial invasion (OR 3.1, 95% CI 1.4–8.1) and invasion depth of >5 mm (OR 2.0, 95% CI: 1.4–2.9) remained significant risk factors for lymph node involvement.

## 4. Discussion

Our results show that the FIGO 2018 staging system accurately reflects the prognosis of patients with early-stage cervical cancer. Comparing the preoperative staging according to FIGO 2009 and FIGO 2018 showed that 501 patients were down-staged from stage IB to stage IA1/IA2. These patients had better prognosis (LNM, parametrial invasion, RFS and OS) than the patients that remained at stage IB1. Furthermore, patients with FIGO 2018 stage IB1 had better prognosis than those with FIGO IB2.

This study showed that the identification of patients with early stage cervical cancer is accurate with the new staging system. Wright et al. also showed an improved discriminatory ability for patients with stage IB tumors [8]. In patients down-staged from FIGO 2009 IB1 to FIGO 2018 IA, the recurrence rate was lower compared to FIGO 2018 IB1 cervical cancer patients. This finding supports the change in staging; however, FIGO 2018 IA1/2 with tumor width > 7 mm had a higher recurrence rate than IA1/2 with tumor width < 7 mm. These groups did not differ in other characteristics and the OS in both groups were comparable.

The most important prognostic factor in cervical cancer is the presence of LNM. Risk factors for lymph node metastases in our study were parametrial invasion, presence of LVSI and invasion depth > 5 mm, which was conform other studies [9,10]. Our study showed LNM in 13.3% of patients who underwent lymphadenectomy. This percentage lies within the range described in the literature (12.2–29.8% in FIGO 2009 IB) [11,12,13,14]. The risk of LNM in patients with FIGO 2018 stage IA1/2 in our study was 4.8%. Wenzel et al. [7], whose study population partly overlaps with our study, showed an incidence rate of 3.5% in early stage cervical cancer in a population of 170 patients with FIGO 2009 stage IB with ≤5 mm depth of invasion and >7 mm horizontal width. We showed that the presence of LVSI significantly increased the risk of LNM; patients with FIGO 2009 IB tumors down-staged to FIGO 2018 IA with LVSI had LNM in 12.7% of cases compared to 2.1% without LVSI. These analyses, but in a smaller group, were also performed by Wenzel et al. and showed comparable differences (9.6% with LVSI and 1.7% without LVSI) [7].

Other studies confirm LVSI as an independent risk factor and emphasize its role in determining the extent of surgery [7,15,16,17]. Determining the presence of LVSI prior to surgery is therefore of great importance, and if inconclusive a second biopsy is needed.

We also showed that the prognosis of patients with a micro-invasive tumor does not depend on depth of invasion (<3 mm or 3–5 mm) but on the presence or absence of LVSI. Therefore, in addition to the role LVSI plays in the indication for lymphadenectomy, we suggest that staging of micro-invasive tumors should take LVSI into account: stage IA1 for tumors with depth of invasion <5 mm without LVSI, and stage IA2 for tumors with depth of invasion <5 mm with LVSI [7,17]. This subdivision would improve the FIGO staging system by more accurately predicting overall survival within the different FIGO stages.

The main risk factors for parametrial involvement are depth of stromal invasion > 2/3, tumor volume, lymph node metastasis and LVSI [18,19,20]. Furthermore, in our study high grade tumors and increasing age were indicated as risk factors. Parametrial invasion was found in 36 patients, of which two patients (0.4%) were down-staged to FIGO 2018 stage IA2, and 11 patients (4.0%), who remained FIGO 2018 stage IB1. This parametrial involvement was assessed histologically as patients with FIGO 2018 IA1/IA2 were not treated differently than prior to the new staging system. These data support the hypotheses of the SHAPE trial (NCT01658930); that parametrectomy might be safely omitted in early stage cervical cancer. The SHAPE trial is a Randomized Phase III Trial that compares radical hysterectomy and pelvic lymph node dissection with simple hysterectomy and pelvic lymph node dissection in patients with FIGO 2009 stage IA2 and IB1 cervical cancer. The recruitment of patients has been completed, and results are expected in 2023. Parametrectomy is the most technically difficult aspect of radical hysterectomy and is the main cause of postoperative complications [21,22,23]. Omitting parametrectomy decreases intra-operative complications like blood loss, bowel or bladder injury and nerve damage. Furthermore, long term complications like voiding problems may significantly decrease. Our results in combination with the existing literature supports omitting parametrectomy in patients with tumors down-staged from FIGO 2009 IB1 to FIGO 2018 IA1 or IA2 (tumors < 20 mm linear extension and ≤5 mm invasion depth). In these patients simple hysterectomy or simple trachelectomy may be sufficient and lymphadenectomy may even be omitted in case LVSI is absent.

Since the results of LACC-trial by Ramirez et al. comparing the oncological safety of minimal invasive surgery versus abdominal surgery in early-stage cervical cancer, the recommendation from laparoscopic/robotic approach changed to an open approach in patients with FIGO 2009 stage IA2, IB and IIA [24]. The majority of our study population who underwent a radical hysterectomy was part of the study by Wenzel et al., evaluating overall survival (OS) and disease-free survival (DFS) in patients treated with abdominal radical hysterectomy (ARH) and laparoscopic radical hysterectomy (LRH) for early-stage cervical cancer. This retrospective study showed equal oncological outcomes between ARH and LRH for early-stage cervical cancer [25]. Currently, the RACC trial is including patients to investigate the oncologic safety of robot-assisted minimal invasive surgery as compared to standard laparotomy in women with histologically confirmed FIGO stage IB1, IB2 and IIA1 disease. Hopefully these results will give more answers regarding the oncologic safety of robot-assisted radical hysterectomies [26].

Important strengths of the study are the national design and the study population size. In addition, the study population is a representable sample of clinical practice contrary to controlled trial environments. It is also the largest cohort study to date to evaluate the prognosis following the revised FIGO 2018 staging for early stage cervical cancer [5]. Limitations of the study, however, include possible bias in quality of data and in data collection inherent to the design of the study. The proportion of women excluded because of missing histology data may be explained by this. Furthermore, the reallocation of stage was based on preoperative findings as per FIGO 2009 staging guidelines, it does not, however, translate to the current usage of FIGO 2018 with the incorporation of imaging. Prospective studies are needed to further assess the long-term outcomes of the revised staging system.

## 5. Conclusions

The revised FIGO 2018 staging system gives a more accurate reflection of the prognosis of patients with early stage cervical cancer compared to the FIGO 2009 staging system, but subdivision in IA1 or IA2 based on presence (newly suggested IA2) or absence (newly suggested IA1) of LVSI instead of depth of invasion would have improved accuracy. Lymph node assessment is not necessary for the FIGO 2018 IA1 or IA2 stages without LVSI, but mandatory in these tumors with presence of LVSI. Parametrial invasion is very rare in patients down-staged from IB1 to IA1/2 and can therefore be omitted. In the near future, results from the SHAPE trial will show whether this also applies to tumors with FIGO 2018 stage IB1.

## Figures and Tables

**Figure 1 cancers-14-03140-f001:**
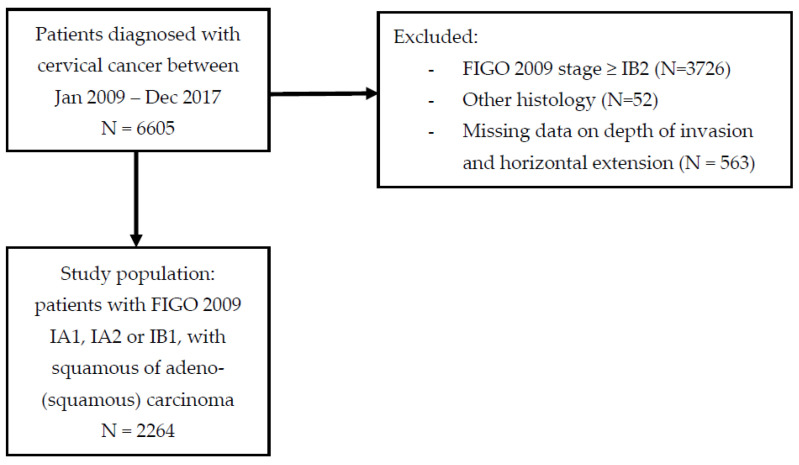
Patient selection flowchart.

**Figure 2 cancers-14-03140-f002:**
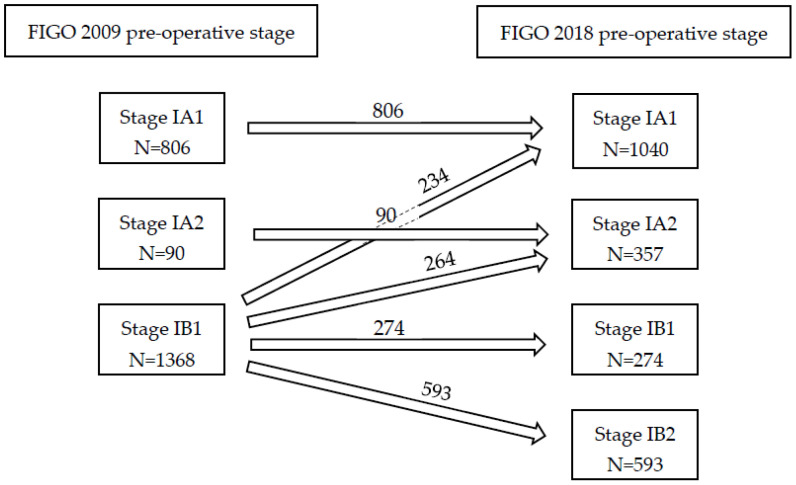
FIGO stage 2009 and allocation to FIGO stage 2018.

**Figure 3 cancers-14-03140-f003:**
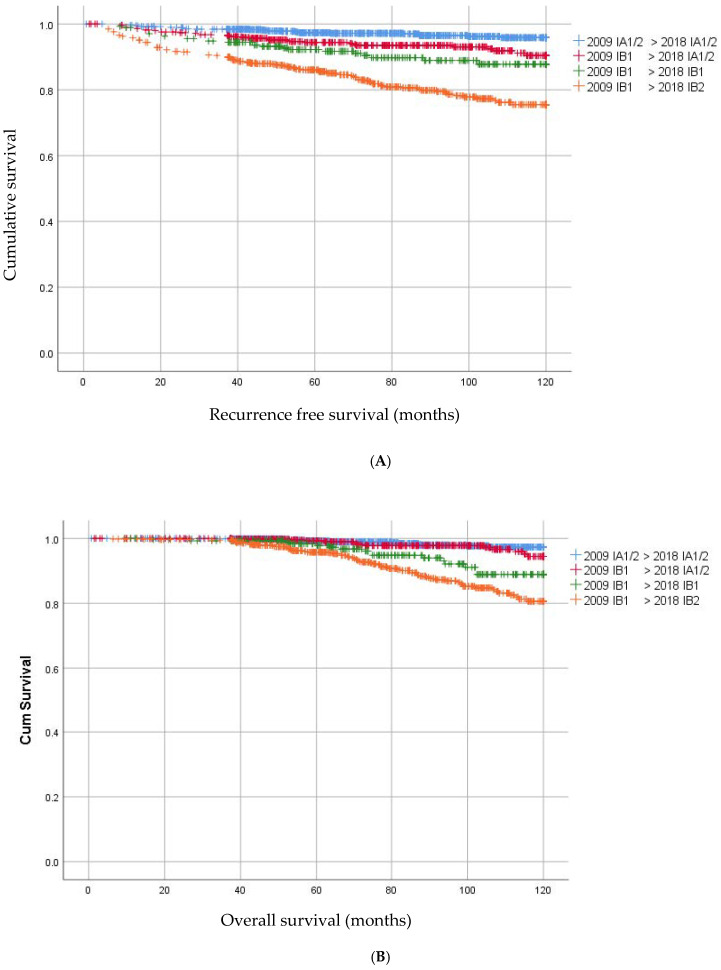
(**A**) Kaplan–Meier curves for overall survival in patients with change in FIGO stage. (**B**) Kaplan–Meier curves for recurrence free survival in patients with change in FIGO stage. (**C**) Kaplan–Meier curves for overall survival in patients with IA1/2 without LVSI and IA1/2 with LVSI.

**Figure 4 cancers-14-03140-f004:**
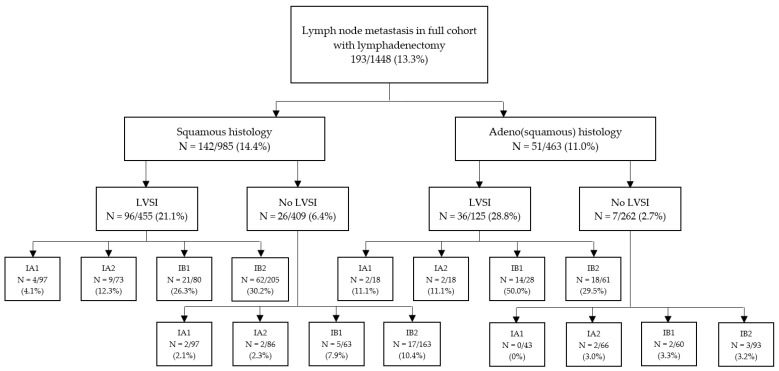
Percentage of patients with lymph node metastasis according to histological characteristics and FIGO stage.

**Table 1 cancers-14-03140-t001:** Comparison of the 2009 and 2018 FIGO staging classifications.

Stage	2009 FIGO Definition	2018 FIGO Definition
I	Confined to the cervix	Confined to the cervix
IA	≤5 mm depth and ≤7 mm width	≤5 mm depth *
IA1	≤3 mm depth	≤3 mm depth
IA2	>3 mm and not >5 mm depth	>3 mm and ≤5 mm depth
IB	>5 mm depth	>5 mm depth
IB1	≤4 cm maximum diameter	≤ 2 cm maximum diameter *
IB2	≥4 cm maximum diameter	>2 cm and ≤4 cm maximum diameter *
IB3	-	>4 cm maximum diameter *
II	Beyond the uterus but not involving the lower one-third of the vagina or pelvic sidewall	Beyond the uterus but not involving the lower one-third of the vagina or pelvic sidewall
IIA	Upper two-thirds of the vagina	Upper two-thirds of the vagina
IIA1	Upper two-thirds of the vagina and ≤4 cm	Upper two-thirds of the vagina and ≤4 cm
IIA2	Upper two-thirds of the vagina and >4 cm	Upper two-thirds of the vagina and >4 cm
IIB	Parametrial invasion	Parametrial invasion
III	Lower vagina, pelvic sidewall and ureters	Lower vagina, pelvic sidewall, ureters and lymph nodes *
IIIA	Lower one-third of the vagina	Lower one-third of the vagina
IIIB	Pelvic side wall	Pelvic side wall
IIIC	-	Pelvic and para-aortic lymph node involvement *
		Pelvic lymph node involvement
IIIC1	-	Para-aortic lymph node involvement
IIIC2	-	
IV	Adjacent and distant organs	Adjacent and distant organs
IVA	Rectal or bladder involvement	Rectal or bladder involvement
IVB	Distant organs outside the pelvis	Distant organs outside the pelvis

* Changes made in the 2018 FIGO staging classification.

**Table 2 cancers-14-03140-t002:** Baseline-, clinical- and histopathology characteristics of study population and stage migration groups.

Characteristics	Total Group N = 2264	IA1/2 = IA1/2 N = 896	IB1 -> IA1/2 N = 501	IB1 = IB1 N = 274	IB1 -> IB2 N = 593	*p*-Value
**Baseline and clinical characteristics**						
Age in years (median IQR)	40 (34–47)	39 (33–45)	40 (34–46)	43 (37–50)	43 (35–51)	<0.05
Body mass index (BMI)						0.43
<18.5 kg/m^2^	39 (1.7%)	7 (0.8%)	13 (2.6%)	11 (4.0%)	8 (1.3%)	
18.5–24.9 kg/m^2^	896 (39.6%)	188 (21.0%)	251 (50.1%)	150 (54.7%)	307 (51.8%)	
25–29.9 kg/m^2^	460 (20.3%)	88 (9.8%)	140 (27.9%	66 (24.1%)	166 (28.0%)	
Obese (≥30 kg/m^2^)	252 (11.1%)	55 (6.1%)	74 (14.8%)	39 (14.2%)	84 (14.2%)	
Unknown	617 (27.3%)	558 (62.3%)	23 (4.6%)	8 (2.9%)	28 (4.7%)	
Surgery						<0.05
Exconisation/LLETZ	439 (19.4)	418 (46.7)	19 (3.8%)	1 (0.4%)	1 (0.2%)	
Simple hysterectomy	496 (21.9)	440 (49.1%)	34 (6.8%)	13 (4.7)	9 (1.5%)	
Radical trachelectomy	145 (6.4)	4 (0.4%)	83 (16.6%)	22 (8.0%)	36 (6.1%)	
Radical hysterectomy	1184 (52.3)	34 (3.8%)	365 (72.9%)	238 (86.9%)	547 (92.2%)	
**Histopathological findings**						
Histology						<0.05
Squamous	1676 (7.0%)	760 (84.8%)	339 (67.7%)	162 (59.1%)	415 (70.0%)	
Adeno	520 (2.0%)	127 (14.2%)	149 (29.7%)	98 (35.8%)	146 (24.6%)	
Adeno-squamous	68 (3.0%)	9 (1.0%)	13 (2.6%)	14 (5.1%)	32 (5.4%)	
Differentiation grade						<0.05
Grade 1	232 (10.2%)	110 (1.3%)	48 (9.6%)	26 (9.5%)	48 (8.1%)	
Grade 2	605 (26.7%)	105 (11.7%)	191 (38.1%)	102 (37.2%)	207 (34.9%)	
Grade 3	380 (1.8%)	26 (2.9%)	89 (17.8%)	72 (2.3%)	193 (32.5%)	
Unknown	1047 (46.2%)	655 (73.1%)	173 (34.5%)	74 (27.0%)	145 (24.5%)	
LVSI						<0.05
Yes	580 (25.6%)	80 (8.9%)	126 (25.1%)	108 (3.,4%)	266 (44.9%)	
No	1038 (45.8%)	367 (41.0%)	292 (58.3%)	123 (44.9%)	256 (43.2%)	
Unknown	646 (28.5%)	449 (5.1%)	83 (16.6%)	43 (15.7%)	71 (12.0%)	
Depth of invasion						<0.05
<3 mm	1058 (46.7%)	807 (90.1%)	235 (46.9%)	4 (1.5%)	12 (2.0%)	
3–5 mm	400 (17.7%)	98 (9.9%)	266 (53.1%)	2 (0.7%)	43 (7.3%)	
>5 mm	794 (35.1%)	0 (0%)	0 (0%)	263 (96.0%)	531 (89.5%)	
Unknown	12 (0.5%)	0 (0%)	0 (0%)	5 (1.8%)	7 (1.2%)	
Positive lymph nodes						<0.05
Yes	193 (8.5%)	1 (0.1%)	24 (4.8%)	48 (17.5%)	120 (20.2%)	
No	1255 (55.4%)	79 (8.8%)	477 (95.2%)	226 (82.5%)	473 (79.8%)	
No lymphadenectomy	816 (36.0%)	816 (91.1%)	0 (0%)	0 (0%)	0 (0%)	
Parametrial invasion						<0.05
Yes	36 (1.6%)	0 (0%)	2 (0.4%)	11 (4.0%)	23 (3.9%)	
No	1293 (57.1%)	38 (4.2%)	446 (89.0%)	(90.9%)	560 (94.4%)	
No parametrectomy	935 (41.3%)	858 (95.8%)	53 (10.6%)	14 (5.1%)	10 (1.7%)	
Follow up in months (median, IQR)	83 (57–111)	84 (58–112)	87 (57–115)	78 (53–106)	80 (56–108)	0.17
Recurrence						<0.05
Total	151 (6.7%)	22 (2.4%)	28 (5.6%)	20 (7.3%)	81 (13.7%)	
Local	78 (3.4%)	17 (1.9%)	19 (3.8%)	6 (2.2%)	36 (6.1%)	
Locoregional	22 (1.0%)	3 (0.3%)	5 (1.0%)	2 (0.7%)	12 (2.0%)	
Distant	51 (2.3%)	2 (0.2%)	4 (0.8%)	12 (4.4%)	33 (5.6%)	
Time to recurrence (months) (median, IQR)	22 (13–45)	26 (15–52)	30 (13–48)	27 (15–46)	19 (10–39)	<0.05

N: number of patients, LLETZ: large loop excision of transformation zone; LVSI: lymphovascular space invasion; IQR: interquartile range.

**Table 3 cancers-14-03140-t003:** Parametrial invasion and clinical characteristics in patients who underwent a parametrectomy.

	Parametrial Invasion N = 36	No Parametrial Invasion N = 1293	*p*-Value Univariate
Age (median, IQR)	54 (range 41–61)	42 (range 35–49)	<0.01
BMI	0 (0%)	31 (2.5%)	0.69
Underweight (<18.5 kg/m^2^)	21 (60.0%)	668 (53.9%)	
Normal (18.5–24.9 kg/m^2^)	10 (28.6%)	348 (28.1%)	
Overweight (25–29.9 kg/m^2^)	4 (11.4%)	192 (15.4%)	
Obese (≥30 kg/m^2^)			
Histology			0.01
Adeno(squamous)carcinoma	5 (13.9%)	434 (33.6%)	
Squamous carcinoma	31 (86.1%)	859 (66.4%)	
Differentiation			<0.01
1	1 (3.2%)	117 (12.7%)	
2	9 (29.0%)	476 (51.8%)	
3	21 (67.7%)	326 (35.5%)	
LVSI			<0.01
Yes	23 (76.7%)	469 (41.4%)	
No	7 (23.3%)	663 (58.6%)	
Invasion depth			<0.01
<3 mm	2 (5.6%)	228 (17.8%)	
3–5 mm	0 (0%)	313 (24.4%)	
>5 mm	34 (94.4%)	740 (57.8%)	
Lateral extent			0.56
<7 mm	2 (5.6%)	107 (8.3%)	
7 mm or more	34 (94.4%)	1186 (91.7%)	

N: number of patients; IQR: interquartile range; BMI: body mass index; LVSI: lymphovascular space invasion.

**Table 4 cancers-14-03140-t004:** Lymph node metastasis and clinical characteristics for all patients who underwent lymph node assessment.

	Lymph Node Involvement N = 193	No Lymph Node Involvement N = 1255	*p*-Value Univariate
Age (median, IQR)	42 (35–50)	41 (35–49)	0.42
BMI			0.19
Underweight (<18.5 kg/m^2^)	5 (2.7%)	28 (2.4%)	
Normal (18.5–24.9 kg/m^2^)	109 (58.9%)	636 (53.9%)	
Overweight (25–29.9 kg/m^2^)	53 (28.6%)	329 (27.9%)	
Obese (≥30 kg/m^2^)	18 (9.7%)	187 (15.8%)	
Histology			0.08
Adeno(squamous)carcinoma	51 (26.4%)	412 (32.8%)	
Squamous carcinoma	142 (73.6%)	843 (67.2%)	
Differentiation			<0.01
1	10 (6.6%)	125 (14.0%)	
2	67 (44.1%)	486 (54.4%)	
3	75 (49.3%)	282 (31.6%)	
LVSI			<0.01
Yes	132 (80.0%)	488 (43.3%)	
No	33 (20.0%)	638 (56.7%)	
Invasion depth			<0.01
<3 mm	12 (6.3%)	298 (23.9%)	
3–5 mm	22 (11.6%)	310 (24.9%)	
>5 mm	155 (82.0%)	639 (51.2%)	
Lateral extent			<0.01
<7 mm	7 (3.6%)	152 (12.1%)	
7 or more	186 (96.4%)	1103 (87.9%)	
Parametrial invasion			<0.01
Yes	18 (9.3%)	18 (1.4%)	
No	175 (90.7%)	1237 (98.6%)	

N: number of patients; IQR: interquartile range; BMI: body mass index; LVSI: lymphovascular space invasion.

## Data Availability

All data generated or analyzed during this study are included in this published article.

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
