# Peer review of "Does the New FIGO 2018 Staging System Allow Better Prognostic Differentiation in Early Stage Cervical Cancer? A Dutch Nationwide Cohort Study"

_cancers, 2022, doi:10.3390/cancers14133140_

Round 1

Reviewer 1 Report

Dear Editor,

Thank you for the opportunity to review this excellent, well written manuscript entitled:” Does the new FIGO 2018 staging system allow better prognostic differentiation in early stage cervical cancer? A Dutch nationwide cohort study"

This is a well-designed retrospective cohort study with a stated objective of:” aimed to evaluate the accuracy of the clinical FIGO 2018 staging in predicting the recurrence free survival (RFS) and overall survival (OS) compared to the FIGO 2009 staging for women with clinically early-stage cervical cancer”. 

I do believe that the objectives of the study are very important and will be of interest to the readers of your journal. I specifically like the large sample size and methodology which will help to provide needed information about the new staging system for patients diagnosed with cervical cancer and its clinical implications.

Minor revisions:

1.     It is unclear from the manuscript which specimen was used to extract data on LVI, was it a pre-op biopsy or final specimen. If final specimen was used, this needs to be clearly stated as pre op biopsy may not show the same results and as such, limit its use.

2.     I believe that the section in the discussion re MRI is not needed, and definitely not supported by data presented in the current manuscript. Does it represent the authors opinion about its use? I think the authors should be encouraged to remove it. 

3.     As data regarding LVI is not always available/ reliable before surgery, can the authors comment on the clinical impact of down-grading a significant number of stage 1b1 patients to 1a1, as common clinical guidelines do not suggest lymph-node assessment for microinvasive disease?

4.     Can the authors provide data about the type of surgery ( MIS vs. open) for the cohort as this could change the risk of disease recurrence in this group of patient?

Again, wanted to congratulate the authors for their excellent work,

Sincerely,

Author Response

Answers to the remarks of reviewers

Reviewer 1:

Dear Editor,

Thank you for the opportunity to review this excellent, well written manuscript entitled:” Does the new FIGO 2018 staging system allow better prognostic differentiation in early stage cervical cancer? A Dutch nationwide cohort study"

This is a well-designed retrospective cohort study with a stated objective of:” aimed to evaluate the accuracy of the clinical FIGO 2018 staging in predicting the recurrence free survival (RFS) and overall survival (OS) compared to the FIGO 2009 staging for women with clinically early-stage cervical cancer”. 

I do believe that the objectives of the study are very important and will be of interest to the readers of your journal. I specifically like the large sample size and methodology which will help to provide needed information about the new staging system for patients diagnosed with cervical cancer and its clinical implications.

Minor revisions:

  1. It is unclear from the manuscript which specimen was used to extract data on LVI, was it a pre-op biopsy or final specimen. If final specimen was used, this needs to be clearly stated as pre op biopsy may not show the same results and as such, limit its use.

Dear reviewer. In the Cancer registration database LVSI was considered positive when either pre-operatively and/or post-operatively LVSI was noticed by the pathologist. Unfortunately, we cannot distinguish in how many patients the presence of LVSI was only seen in the post-operative specimen. We do agree with the reviewer that noticing the presence of LVSI pre-operatively is of great importance, because this influences the extent of the operation in micro-invasive cervical cancer.  We added information on LVSI  to the method section (page 3, lines 95-98). Furthermore, we added a paragraph to the discussion regarding possible discrepancies between pre and post-operative LVSI status (page 11, lines 249-250)

  1. I believe that the section in the discussion re MRI is not needed, and definitely not supported by data presented in the current manuscript. Does it represent the authors opinion about its use? I think the authors should be encouraged to remove it. 

Many thanks for your suggestion. We agree with your opinion and we removed the section in the discussion regarding MRI (page 12, lines 293 – 303)

  1. As data regarding LVI is not always available/ reliable before surgery, can the authors comment on the clinical impact of down-grading a significant number of stage 1b1 patients to 1a1, as common clinical guidelines do not suggest lymph-node assessment for microinvasive disease?

Dear reviewer, this is an interesting point of view. We stated that determining the presence of LVSI prior to surgery is of great importance, as this will influence the extent of surgery. (Discussion session page 11  lines 249-250). In case presence or absence of LVSI cannot be determined on the first biopsy, a second larger biopsy may be needed.

  1. Can the authors provide data about the type of surgery ( MIS vs. open) for the cohort as this could change the risk of disease recurrence in this group of patient?

Dear reviewer, a large part of our study population who underwent a radical hysterectomy was also studied in the article of Wenzel et al.: Survival of patients with early-stage cervical cancer after abdominal or laparoscopic radical hysterectomy: a nationwide cohort study and literature review. Wenzel HHB, Smolders RGV, Beltman JJ, Lambrechts S, Trum HW, Yigit R, Zusterzeel PLM, Zweemer RP, Mom CH, Bekkers RLM, Lemmens VEPP, Nijman HW, Van der Aa MA. .Eur J Cancer. 2020 Jul;133:14-21. This study aimed to evaluate overall survival (OS) and disease-free survival (DFS) in patients treated with abdominal radical hysterectomy (ARH) and laparoscopic radical hysterectomy (LRH) for early-stage cervical cancer. Patients diagnosed between 2010 and 2017 with International Federation of Gynecology and Obstetrics (2009) stage IA2 with lymphovascular space invasion, IB1 and IIA1, were identified from the Netherlands Cancer Registry. This retrospective study showed equal oncological outcomes between ARH and LRH for early-stage cervical cancer. We added this information in the Discussion section page 12, lines 280 to 292, and added both the LACC trial as well as this article to the references. 

Reviewer 2 Report

The article is interesting, written very thoroughly, and most importantly, it has many clinical implications.

The only question, or maybe a comment to the discussion. In the context of the described changes in the FIGO classification, did the authors consider the analysis of ESGO and ESMO recommendations for radical hysterectomy performed by  laparoscopy or robot-assisted surgery, especially in the IA2 and IB1 stages of FIGO 2018 ?

Author Response

Answers to the remarks of reviewers

Reviewer 2:

The article is interesting, written very thoroughly, and most importantly, it has many clinical implications.

The only question, or maybe a comment to the discussion. In the context of the described changes in the FIGO classification, did the authors consider the analysis of ESGO and ESMO recommendations for radical hysterectomy performed by laparoscopy or robot-assisted surgery, especially in the IA2 and IB1 stages of FIGO 2018?

Dear reviewer, you indeed address a very interesting discussion point in the treatment of early stage cervical cancer. A large part of our study population who underwent a radical hysterectomy was also studied in the article of Wenzel et al.: Survival of patients with early-stage cervical cancer after abdominal or laparoscopic radical hysterectomy: a nationwide cohort study and literature review. Wenzel HHB, Smolders RGV, Beltman JJ, Lambrechts S, Trum HW, Yigit R, Zusterzeel PLM, Zweemer RP, Mom CH, Bekkers RLM, Lemmens VEPP, Nijman HW, Van der Aa MA. .Eur J Cancer. 2020 Jul;133:14-21. This study aimed to evaluate overall survival (OS) and disease-free survival (DFS) in patients treated with abdominal radical hysterectomy (ARH) and laparoscopic radical hysterectomy (LRH) for early-stage cervical cancer. Patients diagnosed between 2010 and 2017 with International Federation of Gynecology and Obstetrics (2009) stage IA2 with lymphovascular space invasion, IB1 and IIA1, were identified from the Netherlands Cancer Registry. This retrospective study showed equal oncological outcomes between ARH and LRH for early-stage cervical cancer. We added this information in the Discussion section page 12, lines 280 to 292, and added both the LACC trial as well as this article to the references.

The results of the LACC-trial by Ramirez et al. (Minimally Invasive versus Abdominal Radical Hysterectomy for Cervical Cancer. Ramirez PT, Frumovitz M, Pareja R, Lopez A, Vieira M, Ribeiro R, Buda A, Yan X, Shuzhong Y, Chetty N, Isla D, Tamura M, Zhu T, Robledo KP, Gebski V, Asher R, Behan V, Nicklin JL, Coleman RL, Obermair A.N Engl J Med. 2018 Nov 15;379(20):1895-190) changed the recommendation from laparoscopic/robotic approach to open approach in patients with FIGO 2009 stage IA2, IB and IIA. Currently, the RACC trial (Robot-assisted approach to cervical cancer (RACC): an international multi-center, open-label randomized controlled trial. Falconer H, Palsdottir K, Stalberg K, Dahm-Kähler P, Ottander U, Lundin ES, Wijk L, Kimmig R, Jensen PT, Zahl Eriksson AG, Mäenpää J, Persson J, Salehi S.Int J Gynecol Cancer. 2019 Jul;29(6):1072-1076) is including patients to investigate the oncologic safety of robot-assisted minimal invasive surgery as compared to standard laparotomy in women with histologically confirmed FIGO stage IB (IB3 excluded) and IIA1 disease. Hopefully these results will give more answers on oncologic safety of robot-assisted radical hysterectomies.

Please see the paragraph in the Discussion session, where we addressed these issues. Page 12, lines 280-292.